# Low Energy Availability (LEA) and Hypertension in Black Division I Collegiate Athletes: A Novel Pilot Study

**DOI:** 10.3390/sports11040081

**Published:** 2023-04-07

**Authors:** Troy Purdom, Marc Cook, Heather Colleran, Paul Stewart, Lauren San Diego

**Affiliations:** 1Department of Kinesiology, North Carolina Agricultural and Technical State University, Greensboro, NC 27401, USA; 2Department of Family and Consumer Sciences, North Carolina Agricultural and Technical State University, Greensboro, NC 27401, USA; 3Department of Biostatistics, University of North Carolina, Chapel Hill, NC 27599, USA

**Keywords:** RED-S, cardiometabolic disease, nutrient deficiency, relative energy deficiency syndrome, sudden death, micronutrients, blood pressure

## Abstract

The purpose of this study was to investigate the relationship between low energy availability (LEA) and nutritional content with high blood pressure (HBP) in African American Division I athletes. Twenty-three D1 African American pre-season athletes were recruited to participate. HBP was defined as >120 systolic blood pressure (BP) and <80 diastolic BP. Athletes self-reported nutritional intake using a non-consecutive 3-day food recall which was then reviewed by a sports dietitian. LEA was evaluated as total energy intake—total daily energy expenditure (TDEE), which was predicted. Additionally, micronutrients were evaluated. A statistical analysis relied on Spearman correlation (R), standardized mean difference with 95% confidence interval, mean ± SD, and odds ratios (OR). Correlation values were categorized: 0.20–0.39 = low; 0.40–0.69 = moderate; 0.70–1.0 = strong. A moderate relationship was observed between HBP and LEA (*R* = 0.56) with 14/23 having HBP. Of the 14 athletes observed with HBP, 78.5% (11/14) were calorically deficient (−529 ± 695 kcal) with an OR of 7.2. Micronutrient intake deficiencies were ubiquitous among the 23 HBP athletes: poly-unsaturated fatty acid −29.6%; omega-3 −26.0%; iron −46.0%; calcium −25.1%; and sodium −14.2%, amongst others. LEA and micronutrient deficiencies may contribute to HBP in Black D1 athletes, which has been shown to be the most common modifiable risk factor to decrease the risk of sudden cardiac death.

## 1. Introduction

Division I collegiate athletes would seem to be a healthy population: they exercise well above general health recommendations, maintain agreeable body composition status, and are below the age of morbidity onset thresholds [1]. Current health recommendations along with literature indicate that calorie restriction is effective in mitigating cardiovascular dysfunction compared with medication and/or lifestyle interventions [2]. Moreover, hypertension within the physically active is 50% lower than the general population [3], with sedentarism identified as a root cause of chronic disease [4]. However, collegiate athletes and athletes in general are known to suffer from low energy availability (LEA) [5,6,7] prompting the International Olympic Committee on Sports Nutrition (IOC) to publish a consensus statement on relative energy deficiency syndrome, or RED-S. The IOC [8] described a dose-dependent relationship with LEA and metabolic/hormonal dysfunction [9], perpetuating a stress response that can result in hypertension when LEA becomes a chronic condition [3,10,11,12,13]. Currently, there are only two primary research articles using dated methods that evaluate high blood pressure (HBP) in athletes, with one being a longitudinal study across the competitive season [11,13].

The interplay between diet, physical training stress, and disease is apparent with RED-S when considering system-wide physiological dysfunction [5,6,13]. Despite the potential for morbidity, adaptations to the physical stress incurred with high level training typically induce many positive cardiovascular adaptations that include the following: cardiac chamber dimension, myocardial thickness, vascular adaptations, and microvascular angiogenesis, all of which improve cardiovascular function [14,15]. This presents a paradox where some athletes suffer from HBP which is currently reported at a rate of 3% across all athletes with a higher incidence in certain sports, e.g., American football and water sports [3,12]. Furthermore, HBP onset during pre-season has an 85% chance of advancing into hypertension (Stage I) if left untreated [3]. Revised cardiovascular guidelines published in 2018 by the American Heart Association/American College of Cardiology indicate that for every 20 mmHg and 10 mmHg elevation in BP (systolic/diastolic, respectively), risk factors for stroke, heart disease, and other vascular diseases double [16,17]. Even more concerning is that hypertension is the most commonly observed cardiovascular disease within athletic populations [11,18,19]. Forensic studies have shown that 80% of sudden cardiac deaths (SCDs) in athletes occurred as a result of cardiomyopathy, which is directly related to hypertension [19].

A negative calorie balance is shown to reduce hypertension in the general population which can result in the reduction in cardiovascular disease rates [1,20]. However, student athletes are at a heightened cardiovascular disease risk due to the lack of understanding of what conditions (LEA, sleep deprivation, etc.) perpetuate HBP and SCD in what is perceived to be a healthy population. For example, African American Division I basketball athletes have a significantly elevated cardiovascular risk 10× that of the overall athlete population (1:5200 vs. 1:53,703) [21]. The available evidence indicates that LEA in student athletes is associated with general health deterioration, e.g., bone health, metabolic dysfunction, cardiovascular impairment, etc. [22]. It is known that LEA is the result of a daily calorie deficiency (consumed vs. utilized), also known as a negative energy balance, and that system-wide dysfunction occurs that includes cardiovascular disease if chronic [5,12,23]. Energy deficiencies among collegiate athletes are well-documented, with one study showing an average calorie deficiency of −595 ± 283 kcals/day (95% CI: −878, −312 kcals/day) [24]. Despite the resources to compete at elite levels, student athletes are consistently deprived of sufficient calories while competing, which results in adverse effects beyond performance [5,7,22,25] that include illness, hormonal fluctuations, and increased resting heart rate amongst others [22,24,25]. In addition, the NIH in 2008 was compelled to write protocols to screen for heart disease to safeguard athletes from cardiovascular disease risk [26]. Despite relevant concerns, the current literature separately provides evidence of the consequences of HBP in athletes and LEA as a concern in the same population. Notwithstanding the significant body of evidence that separately evaluates the two issues in competitive athletes (LEA and HBP), presently, only two primary articles exist that document hypertension in athletes [11,13] and neither consider nutritional deficiencies despite the 2018 IOC on Sport Nutrition warning of the perceived risks [5,6]. Therefore, the primary aim for this study was to investigate the relationship between LEA and HBP in Division I collegiate athletes. Additionally, we hypothesized an association with nutritional content when HBP was observed.

## 2. Materials and Methods

### 2.1. Participants

Participants were recruited by word of mouth amongst a Division I Historically Black College or University (HBCU) population of >300 athletes who were cleared to play after an annual pre-participation health screen/physical. Thirteen males and ten female healthy Black Division I collegiate athletes (*n* = 23) (age: 19.7 ± 1.5 yrs; height: 178.5 ± 12.5 cm; mass: 77.9 ± 19.1 kg; fat free mass: 62.2 ± 12.5 kg; fat mass: 15.7 ± 10.2 kg; body fat percent: 19.2 + 8.2%; VO_2_: 54.5 ± 12.1 mL/kg/min) initially provided informed written consent to participate in this study that was approved according to the Declaration of Helsinki by the University Review Board (IRB#17-0148). These 23 athletes represented a variety of sport backgrounds that included volleyball (*n* = 5), track (*n* = 9), American football (*n* = 1), and men’s basketball (*n* = 8). After providing informed written consent, the 23 pre-season athletes were enrolled and evaluated.

Height was measured with a wall-mounted stadiometer (Seca Model 216, Hanover, MD, USA) along with weight and body composition via 8-point bioelectrical impedance analysis (Seca, Model MBCA514, Hanover, MD, USA). VO_2_ max via the Bruce protocol treadmill stress test was used to evaluate cardiorespiratory fitness. Expired gases were measured using a metabolic cart (Parvomedics True 2400). Resting energy expenditure was evaluated using the Shofield Equation according to the World Health Organization Joint Expert Consultation on Human Energy Requirements [24,27,28] with a standardized physical activity level of 1.8 relevant to the pre-season condition [28]. All athletes rested sitting for >5 min before their BP was measured according to ACSM standardized guidelines [1]. High BP (HBP) was defined according to the 2018 American College of Cardiology/American Heart Association Executive Summary: systolic BP > 120 mmHg and/or diastolic BP > 80 mmHg [16,17]. HBP was further classified as Stage I hypertension if systolic < 139 or diastolic < 89; else, Stage II.

### 2.2. Dietary Recall

After the laboratory evaluations were completed, participants completed a food recall interview to assess their nutritional intake using the Nutritional Data System for Research (NDSR) [29,30,31]. The NDSR has been previously validated using doubly labeled water and proved to be an accurate method to assess energy intake [29,31]. The NDSR uses a three-day non-consecutive food recall that includes two weekdays and one weekend day over a week-long period. After logging their daily nutritional intake which included the time of day, food content, and the amount of food they consumed per item, a sports dietitian performed a 5-step dietary review of their food records and clarified the nutritional food logs. The 5-step dietary review increased the specificity, accuracy, and reproducibility of the energy and nutrient intake while reducing bias [29,31] amongst the food logs reviewed.

### 2.3. Low Energy Balance (LEA) Assessment

Energy balance was defined as the equitable contribution of energy intake and energy expenditure.
Energy balance = (energy expenditure − energy intake)

Energy and nutrient intake compositions were quantified using the NDSR to assess LEA while energy expenditure was estimated as described above. LEA was defined as an athlete consuming less energy than required for their estimated daily energy expenditure and quantified as a negative energy balance. When athletes do not consume enough energy to meet their energy expenditure demands, LEA is present. When LEA becomes a chronic condition, athletes can experience a deleterious systematic function [5,6,24] known as relative energy deficiency syndrome (RED-S), perpetuating a stress response that can induce HTN [3,5,10,12].

### 2.4. Statistical Analysis

The statistical analysis strategy focused on descriptive and exploratory measures of association and differences. The small sample size could not support additional model-based methods such as covariate adjustment (e.g., for the various sports) or model-based methods that might account for confounding factors known to influence both HBP and LEA. Consequently, inferential methods such as hypothesis testing and use of p-values were not appropriate. This report relied on Spearman correlation coefficients, standardized mean differences with pooled estimates of variance (as suggested by Cohen [32]), 95% confidence interval estimates, and the odds ratio (OR) for the relationship between two binary variables: P_1_ = caloric deficiency (yes, no) and P_2_ = HBP (yes, no).

## 3. Results

Prevalence of HBP was greater among the 14 participants with LEA, P_1_ = 78.6% [CI: 57%, 100%] relative to the 9 participants without LEA, P_2_ = 22.2%, [CI: 0%, 49%]. Individual comparisons are shown within Figure 1. Estimates of the difference between the two proportions were P_1_ − P_2_ = 56.4% [22%, 91%]. The estimate of the Spearman correlation between P_1_ and P_2_ was R = 0.56 [0.21, 0.90] (Table 1). These data are consistent with the suggestion that those with LEA may be 12 times more likely to experience HBP (OR = 12.8).

Micronutrient profiles for the athletes with HBP and caloric deficiency are shown in Table 2. All nutrients (macronutrients and micronutrients) were resoundingly deficient in the 13 athletes with HBP compared to the 10 without HBP.

## 4. Discussion

The purpose of this observational study was to investigate the relationship between LEA and nutritional intake on BP in Black Division I athletes. To our knowledge, this is the first study to evaluate the nutritional influence on HBP and separately the first to consider Black athletes. Previous research that did not consider race has shown a 3% prevalence of HBP [3,12] in athletes with an 85% chance that following diagnosis, HBP will worsen over the course of the competitive season [3]. While our sample was assessed during the pre-season, 14/23 or 56% suffered from HBP, indicating a moderate relationship (Figure 1, Table 1). However, when further evaluating the 14/23 athletes who suffered from HBP, 1l/14 or 78.6% were calorically deficient with an average deficiency of −529 ± 695 kcal that was mirrored by the deficient micronutrient intake.

The IOC on Sports Nutrition [5] describes a dose-dependent relationship with LEA and metabolic/hormonal dysfunction [9] perpetuating a stress response that can result in cardiovascular disease as a chronic condition [3,10,11,12] known as RED-S [5]. In our sample, 10/11 who had adequate energy availability met the IOC’s recommendations of >45 kcal/kg FFM/day, while 8/12 of those who were hypertensive did not meet the IOC’s recommendations [5]. Interestingly, the IOC acknowledges a universal absence of minorities within the reported literature, making this a novel study and expressing the need for more rigorous studies investigating nutrition and cardiovascular disease in minority athletes. For instance, student athletes, in general, experience a heightened risk of RED-S and/or cardiovascular disease due to the lack of understanding of what conditions perpetuate HBP [21], which our results support. The risks of physiological dysfunction, that include cardiovascular are further compounded due to a lack of understanding of nutrient needs among the athletic population that leads to false perceptions, unhealthy eating behaviors [24,34,35], and an inverse relationship with nutrition knowledge and body fat [35]. Student athletes are consistently deprived of adequate energy while maintaining high physical workloads that result in adverse effects that include cardiovascular dysfunction, illness, endocrine dysfunction including amenorrhea and bone density depletion, amongst others [5,14,22,24,25]. Physiological dysfunction is further compounded by the inadequate consumption of macronutrients, which were universally deficient in those with high blood pressure (Table 2). However, macronutrient deficiencies were likely to occur with LEA considering those with HBP in our sample maintained an average deficiency of −529 ± 695 kcal, while the normotensive had a positive energy balance of +624 ± 1120 kcal: a 1163 kcal difference. Our results indicate that 86% of those with LEA suffered from HBP exacerbating morbidity when compared to the previous literature that reports a 3% prevalence of hypertension amongst athletes [3,12].

An athlete’s cardiovascular adaptation to exercise stress during the competitive season is generally positive. Cardiovascular adaptations as a result of training include increased heart chamber dimension, heart wall thickness, and increased microvasculature, all of which improve cardiovascular performance including BP regulation [14,15]. Paradoxically, despite the vast network of positive cardiovascular adaptations to exercise, athletes in our sample still exhibited HBP that commonly go undiagnosed [21,26] and is further complicated by family history and body composition changes [11,12,35]. However, throughout the competitive season, athletes are known to express higher levels of psychological stress and endure high frequencies of long-duration high intensity exercise paired with travel and academic expectations [11,24] that can compromise cardiovascular health [22]. LEA and frequent competitions can increase the allosteric load that can negatively resonate as elevated BP if it becomes a chronic condition [11,22], which can be described as RED-S but has yet to be observed within the literature. Hypertension is the most observed cardiovascular disease within athletic populations [3,12], while those with elevated BP were shown to have a 1.7× higher risk for development of hypertension in a five year follow-up [36]. Furthermore, in 2008, the NIH developed a screening protocol with recommendations to evaluate athletes for pre-participation heart disease to reduce the risk of sudden death in athletes [26], while 80% of all SCD in athletes are the result of cardiomyopathy which is directly related to hypertension [19]. The current screening recommendations include pre-participation health screening, while the competitive season has been shown to exacerbate HBP in American-style football collegiate athletes [11]. Therefore, RED-S may contribute to the heightened risk of cardiac events despite the current screening methods [26] that often go unchecked after pre-participation screens, of which our results agree (Figure 1).

Micronutrients are known to affect cardiovascular behavior due to underconsumption, overconsumption, and/or lack of diversity with dietary intake [7]. Moreover, due to LEA and the inherent activities and environments in which athletes participate, athletes are prone to nutrient deficiencies that can elevate the risk of HBP [5,7,22,25]. Known micronutrients that influence hemodynamics include but are not limited to the following: sodium, calcium, omega-3 fatty acids, and iron, amongst others. In this study, the micronutrient consumption of those with HBP was uniformly deficient when compared to normotensive athletes (Table 2). This is not surprising in that LEA is inherently nutrient-deficient, and therefore the opportunity to consume adequate nutrients is diminished. While LEA is shown to be common in athletes [5,24,25], the moderate relationship with LEA and HBP (Table 1) reinforces the concern originally published by the IOC [5,8] where 78.5% percent of Black D1 athletes with LEA in this sample expressed cardiovascular morbidity (Figure 1). These findings further indicate that the previously reported HBP at a rate of 3% amongst athletes may be underestimating morbidity rates among athletes when considering the 2018 Executive Summary revising the BP guidelines from the American College of Cardiology/American Heart Association on Clinical Practice Guidelines [16,17] that could be further exacerbated by LEA.

The diversification of dietary intake that includes adequate quantities of micronutrients is integral to support both performance and athlete health [5,7,26]. Moreover, independent of energy intake, micronutrient deficiencies have been shown to affect cardiometabolic function: (1) calcium, necessary for muscle contraction that includes cardiac and smooth muscle control was 25.1% deficient in those with HBP (Table 2). Previous studies indicate that deficiencies in calcium elevate hemodynamic function through mechanisms that activate vasoconstriction [7,37], including intracellular smooth muscle vasculature calcium concentrations, parathyroid function, and activation of the renin-angiotensin system [37]. (2) Vitamin D intake was 32% deficient in those with HBP compared to the normotensive which was shown to increase cardiovascular risk factors [38,39] that include incidence of hypertension and/or SCD [38]. Adequate vitamin D intake is also known to affect calcium absorption downregulating the renin-angiotensin system by inhibiting renin production and improving arterial compliance [38]. (3) Omega-3 FAs are shown to be critical for cardiovascular and neurologic development [40,41]. A recent study of >400 D1 collegiate football players indicated that all athletes evaluated were considered to have an intermediate or high risk of cardiovascular disease [40]. Our sample echoed their findings using the Omega-3 index (Omega-3 index = Omega-3 FAs ÷ total fat × 100) where 90% of the athletes in our sample, regardless of hemostatic function were considered high risk (<4% total fat intake) using the same criteria. However, the Omega-3 deficiency contribution to HBP in our sample is further explained by the 23.4% deficiency compared to the normotensive. (5) Iron is a necessary micronutrient shown to affect cardiovascular function contributing to the formation of hemoglobin for O_2_ transport. Athletes are known to suffer from iron deficiency as a result of their training due to losses from mechanical hemolysis, intestinal bleeding, hematuria, sweating, and poor intestinal absorption which can promote anemia [42,43]. Furthermore, iron deficiencies can occur during strenuous physical activity and energy deficits due to the overproduction of hepcidin, a regulatory iron-absorbing hormone [43]. In our sample, those with HBP consumed 54% iron when compared to the normotensive, which can negatively influence cardiometabolic function in those who cannot supplement iron loss as a result of high training loads paired with a negative energy balance (LEA).

The strengths of this study are that it is the first of its kind to investigate nutritional factors and their relationship to HBP in Black D1 athletes. However, this study has several limitations. First, all subjects were African American which limits generalizability. Second, the small sample size does not allow us to address the imbalanced representation of the various sports and other characteristics. Third, the small sample size does not allow us to address biases due to confounding factors; e.g., psychological stress, lack of sleep, obesity, alcohol, and nicotine are known to affect appetite and hemodynamic function. These and other confounders went unmeasured in this study and should be addressed in future investigations to more accurately assess the relationship between LEA and HBP. Fourth, as a consequence of those limitations the data may provide evidence of association but cannot support conclusions about cause and effect.

## 5. Conclusions

In summary, we observed a moderate association between LEA and HBP in Black Division I athletes, thus suggesting a relationship with cardiovascular disease and RED-S. While this observational study was conducted with a small sample of Black Division I athletes (*n* = 23), it is the first to consider RED-S and nutrient deficiencies that predicate cardiovascular dysfunction first published by the IOC. Beyond performance decrements, the caloric deficits athletes experience have deleterious physiological consequences when chronic, that include observed dysfunction in cardiovascular, endocrine, renal, reproductive, skeletal, and muscular systems. Even more concerning is that LEA can significantly increase the risk of hypertension onset which may lead to SCD. The cardiovascular dysfunction that athletes experience can promote lifelong morbidity. Educating athletes and their support staff on how to meet energy/nutrient intake recommendations while increasing nutritional awareness can reduce the consequences of nutrition mismanagement and positively influence both the heath and performance of the athletic community.

## Figures and Tables

**Figure 1 sports-11-00081-f001:**
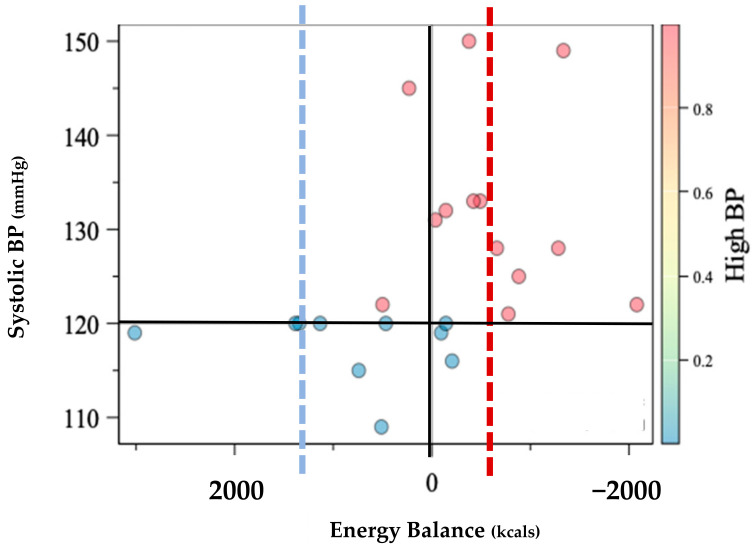
Individual distribution of systolic blood pressure (BP) according to energy balance assessment in kilocalories (Kcals). Energy balance was defined as equitable energy expenditure and energy intake. Dotted lines represent the mean energy intake of normotensive group (*n* = 10) in blue while the high blood pressure group mean (*n* = 13) is shown in red.

**Table 1 sports-11-00081-t001:** Spearman correlation (*n* = 23) according to the two dichotomous variables, high blood pressure [15,16,33] and caloric deficiency (energy intake < energy expenditure) [5,6,23]. In addition, point estimates are calculated according to (yes, no) as a percentage with the corresponding 95% confidence intervals (95% CI).

		Blood Pressure			Point Estimate	95% CI
		High	Normal	Total		
** Caloric Deficiency **	Yes	11	3	14	P_1_ = 0.79	[0.57, 1.00]
	No	2	7	9	P_2_ = 0.22	[0.00, 0.49]
	Total	13	10	23	P_1_ − P_2_ = 0.564	[0.22, 0.91]
	Spearman Correlation Coefficient				R = 0.56)	[0.21, 0.90]

**Table 2 sports-11-00081-t002:** Group comparisons according to the binary variables high blood pressure (HBP) and caloric deficiency (yes vs. no). Values are expressed as group means, minimum (min) and maximum (max) recorded value, and standard deviation (SD). (TEE): predicted total energy expenditure; (kcal): kilocalorie; (mmHg): milometers of mercury; (g): grams; (PUFA): polyunsaturated fatty acid; (Omega3): omega three fatty acid; (mg) milligram.

ENERGY BALANCE AND BLOOD PRESSURE			
**Variable**	**HBP**	**Mean**	SD
**TEE** (kcal)	N	2952.5	404
	Y	3336.4	716
**TEI** (kcal)	N	3767.5	1158
	Y	2739.4	1087.0
**Systolic BP** (mmHg)	N	117.8	3.6
	Y	132.2	10.0
**Diastolic BP** (mmHg)	N	74.0	4.2
	Y	81.9	6.6
MACRONUTRIENTS			
**Carbohydrate** (g)	N	499.2	202.6
	Y	344.7	126.6
**Protein** (g)	N	135.4	34.5
	Y	109	57.1
**Lipids** (g)	N	143	42.8
	Y	106.2	44.4
MICRONUTRIENTS			
**PUFA** (g)	N	40.7	14.6
	Y	28.8	15.1
**Omega3** (g)	N	3.7	1.0
	Y	2.9	1.7
**Sodium** (mg)	N	5750	1759
	Y	4938	2470
**Calcium** (mg)	N	1436	536
	Y	1077	589
**Vitamin D** (mg)	N	7.7	7.6
	Y	5.2	5.2
**Iron** (mg)	N	28.4	15.3
	Y	15.4	6.0

## Data Availability

Not applicable.

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
