# Peer review of "Low Energy Availability (LEA) and Hypertension in Black Division I Collegiate Athletes: A Novel Pilot Study"

_sports, 2023, doi:10.3390/sports11040081_

Round 1

Reviewer 1 Report

Thank you for the opportunity to revise the present MS. This is a well written MS, and I only have some comments for Methods section that need Please find my comments and suggestions below.

Was there any a priori sample size estimation ? This would give a reader a better insight ...

Since you tested female participants were you taking cycle phase into consideration. This could strongly affect your results. Please read and try to implement in your work 

Myths and Methodologies: Reducing scientific design ambiguity in studies comparing sexes and/or menstrual cycle phases - PubMed (nih.gov)

Typo, VO2 max should be presented as V̇O2, as this is the correct way to indicate flow, here and in the MS.

Please describe the CPET test to measure VO2 max. ? Add data on metabolic analyzer used in your work...

Line 165 - why are these data in yellow

And for the conclusion, have you considered the role of individual cardiorespiratory fitness in BP regulation ?

Acute flywheel exercise does not impair the brachial artery vasodilation in healthy men of varying aerobic fitness - PubMed (nih.gov)

Author Response

  1. Was there any a priori sample size estimation? This would give a reader a better insight ...

RESPONSE: Thank you for this comment. The authors agree that a priori power calculation is an important component of research design. However, no priori sample size was conducted as this is/was a preliminary pilot study to assess if there was a relationship between BP and dietary intake among athletes with and without high BP. No data exist in this area making it difficult to base any comparisons. Furthermore, the homogenous population of minority athletes further suggests consideration as an important contribution to the field that the IOC (international Olympic Committee) acknowledges that no data exist in this population (Maugans et al. 2019). Additionally, we detail both the cross-sectional relationships as well as the limitations within the methods, results, and discussion/limitation sections.

  1. Since you tested female participants were you taking cycle phase into consideration. This could strongly affect your results. Please read and try to implement in your work Myths and Methodologies: Reducing scientific design ambiguity in studies comparing sexes and/or menstrual cycle phases - PubMed (nih.gov)

RESPONSE: Thank you for this comment. The authors agree that female agency in research is lacking. However, as sex-based differences is not a primary aim of this study we did not account for menstrual cycle phase or include it within our analysis. We did however confirm that no participants were on their cycle prior to testing. Further, while female reproductive cycles are shown to have physiological effects, current evidence does not indicate relevant magnitude exist regarding sex based differences in resting hemodynamic function (citations below), which our study highlights. However, we have included language within our limitations that express this concept such as: “As well, confounding factors that could incrementally increase BP e.g.: sex-based differences, psychological stress, lack of sleep, obesity, and alcohol are known to affect appetite and hemodynamic function and should be investigated with more rigorous research designs.”

https://link.springer.com/article/10.1007/s00421-021-04656-x

https://www.frontiersin.org/articles/10.3389/fphys.2021.654585/full

  1. Typo, VO2 max should be presented as V̇O2, as this is the correct way to indicate flow, here and in the MS.

RESPONSE: All nomenclature has been changed to reflect V̇O2 throughout the manuscript.

  1. Please describe the CPET test to measure VO2 max. ? Add data on metabolic analyzer used in your work...

RESPONSE: information about CRF testing has now been added to the manuscript. Specifically, “Participants then completed a V̇O2 test to evaluate cardiorespiratory fitness using the Bruce Protocol. Expired gases were measured using (Parvomedics True 2400).”

  1. Line 165 - why are these data in yellow

RESPONSE: we apologize for this oversight. Highlights have been removed

  1. And for the conclusion, have you considered the role of individual cardiorespiratory fitness in BP regulation ? Acute flywheel exercise does not impair the brachial artery vasodilation in healthy men of varying aerobic fitness - PubMed (nih.gov)

RESPONSE: The authors do consider CRF as a determinant for BP function and acknowledge the vast amount of literature to support this relationship. As such, we report the average V̇O2 of the cohort were in the 85th percentile in the methods section (specifically “participants”) or classified as excellent according to ACSM Guidelines 11th Ed. The authors account for this relationship within our research design by working to minimize CRF as a variable, which is why we specifically investigated homogenous sample of Division I athletes and included CRF as a demographic variable as it relates to cardiovascular health. We further elaborate on this specific concept within the introduction, “Current health recommendations along with literature support indicate that calorie restriction is effective in mitigating cardiovascular dysfunction compared with medication and/or lifestyle interventions [2]. Moreover, hypertension within the physically active is 50% lower than the general population [3] with sedentarism identified as a root cause of chronic disease [4].”  AND  “Despite the potential for morbidity, adaptations to the physical stress incurred with high level training typically induces many positive cardiovascular adaptations that include: cardiac chamber dimension, myocardial thickness, vascular adaptations, and microvascular angiogenesis all of which improve cardiovascular function [15,16]. This presents a paradox where some athletes suffer from HTN that is currently reported at a rate of 3% across all athletes”

We hope our responses have satisfied the reviewer’s relevant concerns about study design, consideration of confounding factors, and value of the novel results.

Reviewer 2 Report

- Describe the novelty of the study more clearly.

- Add a procedure subheading describing how the authors got in contact with potential participants by also writing the study design and sampling method.

- Which test was used to examine the mean difference between sex in the results disclosed in Table 1? please clarify.

- provide sample size calculations for transparency indicating which software or research was used as a reference. A power analysis calculator typically requires you to input the desired power level, the effect size, the significance level, and the variability.

- Tables should follow journal style.

- Add notes for each table for clarity.

-

Author Response

  1. Describe the novelty of the study more clearly.

RESPONSE: Thank you for your comment. The authors have included statements to support the novelty of the study throughout the introduction. The statements below are obtained from each paragraph throughout the introduction including the justification statement prior to purpose statement, which is the final statement shown below. The authors are happy to oblige reviewer specific comments to enhance the manuscript. Comments referenced above are shown below: “The IOC [8] describe a dose-dependent relationship with LEA and metabolic/hormonal dysfunction [9] perpetuating a stress response that can result in hypertension when LEA becomes a chronic condition [3,10–13].” AND   “Despite the potential for morbidity, adaptations to the physical stress incurred with high level training typically induces many positive cardiovascular adaptations that include: cardiac chamber dimension, myocardial thickness, vascular adaptations, and microvascular angiogenesis all of which improve cardiovascular function [15,16]. This presents a paradox where some athletes suffer from HTN that is currently reported at a rate of 3% across all athletes with higher incidences in certain sports e.g.: American football and water sports [3,12].”  AND    “A negative calorie balance is shown to reduce hypertension in the general population which can result in the reduction of cardiovascular disease rates [1,21]. However, student athletes are at heightened cardiovascular disease risk due to the lack of understanding what conditions (LEA, sleep deprivation, etc) perpetuate HBP and SCD in what is perceived to be a healthy population. For example, African American Division I basketball athletes have a significantly elevated cardiovascular risk 10x that of the overall athlete population (1:5200 vs 1:53,703) [22].”   AND   “Notwithstanding the significant body of evidence that separately evaluate the two issues in competitive athletes (LEA and HBP), presently, only two primary articles exist that document hypertension in athletes [11,13] and neither consider nutritional deficiencies despite the 2018 IOC on Sport Nutrition warning of the perceived risks [5,6]. Therefore, the primary aim for this study was to investigate the relationship between LEA and HBP in Division I collegiate athletes.

  1. Add a procedure subheading describing how the authors got in contact with potential participants by also writing the study design and sampling method.

REPONSE: Thank you for this observation and suggestion. The authors have added language to describe subject recruitment that include: “Participants were randomly recruited by word of mouth amongst a Division I Historically Black College or University (HBCU) population of >300 athletes who were cleared to play after an annual pre-participation health screen/physical.”

  1. Which test was used to examine the mean difference between sex in the results disclosed in Table 1? please clarify.

RESPONSE: Thank you for this comment. T-tests were used to evaluate sex-based differences. However, after reflecting upon the reviewer comments and considering sex-based comparison were not included within the study design reflective of the pilot nature of study, the authors have removed Table I entirely and included group demographic means + SD with N size.

  1. provide sample size calculations for transparency indicating which software or research was used as a reference. A power analysis calculator typically requires you to input the desired power level, the effect size, the significance level, and the variability.

REPSONSE: Thank you for this comment. The authors agree that a power analysis is an important component of research design. However, as this is/was a preliminary pilot study to assess if there was a relationship between BP and dietary intake among athletes with and without high BP. We also would like to reiterate that no data exist that consider the relationship between cardiovascular dysfunction and nutritional deficiencies in athletes. Furthermore, this study represents the only data set to consider minority focused representation within the literature regarding this topic despite black athletes and/or mixed race athletes comprising of >21% of all collegiate athletes (statistics sourced from NCAA data base 3/28/23). As well, the data from this study are being used to justify further research in the area and is warranted to consider cross-sectional study designs as well as longitudinal studies.https://www.ncaa.org/sports/2018/12/13/ncaa-demographics-database.aspx

  1. Tables should follow journal style.

RESPONSE: Thank you for bringing this to our attention. All tables and figures have been formatted to author guidelines.

  1. Add notes for each table for clarity.

RESPONSE: all table descriptions have been carefully considered for clarity and revised as necessary.

Round 2

Reviewer 2 Report

No further comments.